# WNT-FRIZZLED-LRP5/6 Signaling Mediates Posterior Fate and Proliferation during Planarian Regeneration

**DOI:** 10.3390/genes12010101

**Published:** 2021-01-15

**Authors:** Eudald Pascual-Carreras, Miquel Sureda-Gómez, Ramon Barrull-Mascaró, Natàlia Jordà, Maria Gelabert, Pablo Coronel-Córdoba, Emili Saló, Teresa Adell

**Affiliations:** 1Department of Genetics, Microbiology and Statistics and Institute of Biomedicine, Universitat de Barcelona, 08028 Barcelona, Catalunya, Spain; eudald.pascual@ub.edu (E.P.-C.); msureda@fbg.ub.edu (M.S.-G.); ramonbbm@gmail.com (R.B.-M.); nataliajorda6@gmail.com (N.J.); mariagelabert_611@hotmail.com (M.G.); pablocoronel14@gmail.com (P.C.-C.); esalo@ub.edu (E.S.); 2Institut de Biomedicina de la Universitat de Barcelona (IBUB), Universitat de Barcelona, 08028 Barcelona, Catalunya, Spain

**Keywords:** Wnt pathway, Wnt receptors, posterior fate, proliferation, regeneration, planarians

## Abstract

An organizer is defined as a group of cells that secrete extracellular proteins that specify the fate of surrounding cells according to their concentration. Their function during embryogenesis is key in patterning new growing tissues. Although organizers should also participate in adult development when new structures are regenerated, their presence in adults has only been identified in a few species with striking regenerative abilities, such as planarians. Planarians provide a unique model to understand the function of adult organizers, since the presence of adult pluripotent stem cells provides them with the ability to regenerate any body part. Previous studies have shown that the differential activation of the WNT/β-catenin signal in each wound is fundamental to establish an anterior or a posterior organizer in the corresponding wound. Here, we identify the receptors that mediate the WNT/β-catenin signal in posterior-facing wounds. We found that Wnt1-Fzd1-LRP5/6 signaling is evolutionarily conserved in executing a WNT/β-catenin signal to specify cell fate and to trigger a proliferative response. Our data allow a better understanding of the mechanism through which organizers signal to a “competent” field of cells and integrate the patterning and growth required during de novo formation of organs and tissues.

## 1. Introduction

An organizer can be defined as a group of cells that secrete an extracellular protein that can specify the fate of the surrounding cells according to their concentration [1,2,3,4]. A broader vision of organizers includes other essential properties: (i) the receiving tissue must be “competent” to receive the signal, and (ii) specifying a pattern must come together with promoting growth in order to form a new and complete patterned structure [1]. Thus, organizers pattern a field of cells which is in continuous growth. The term organizer was used for the first time by Spemann and Mangold, when they discovered the ability of dorsal cells in the blastopore lip of amphibians to induce the formation of a complete axis when grafted to the opposite site of a second embryo [5]. The homologous structure has also been identified in all vertebrates and has been given different names, such as Hensen’s node in the chick [6]. Organizing centers, referring to organizing cells that pattern a tissue or an organ but not a complete body axis, have been identified in several stages of development of all organisms, for instance, in the limb bud of tetrapods [7] or the isthmic organizer at the midbrain–hindbrain boundary [8]. As observed, organizers or signaling centers have been mainly studied in embryonic stages. However, the first experiment that suggested their existence was performed in 1909 by Ethel Browne using adult Hydra. By transplanting head tissue into the body column of a host, she observed the induction of a secondary axis that was predominantly made of host cells, concluding that the *Hydra* head has the ability to instruct the fate of the host tissue [9].

Although organizers should also participate in adult development (for instance, when new structures must be regenerated), the presence of organizers in adult tissues has received little attention since its discovery in 1909 [2], and their presence has only been identified in few species with striking regenerative abilities. Molecular and genetic studies in whole-body-regenerating animals, such as *Hydra* and planarians, have demonstrated that the formation of organizers in the wounds is a crucial step to specify the fate of the regenerating tissue [2,10]. In zebrafish, organizing cells have also been identified in the tip of the regenerating caudal fin [11]. All of these studies have identified the WNT/β-catenin (cWNT) as the key signal providing organizing properties. Thus, the cWNT signaling pathway appears as an evolutionary conserved mechanism to specify the primary body axis both during regeneration and during embryogenesis in all metazoans [12,13,14].

Planarians provide a unique model system to understand the function of organizers during adult development and, more specifically, to study the properties of the organizing and the receiving cells in the “competent” tissue and its implication in the growth associated with regeneration. This is because planarians can regenerate any body part due to the presence of a huge population of adult pluripotent stem cells, called neoblasts [15,16], and because they are Lophotrochozoa [17], which show bilateral symmetry and cephalization; thus, their regeneration includes complex organs. After amputation, an apoptotic and a proliferative response takes place, which should be coordinated to allow proper regeneration and restoration of the missing structures [18,19]. Muscle cells in the regenerative tips act as organizers, secreting molecules which will pattern the regenerating blastema according to the pre-existent tissue [20,21]. As mentioned, the cWNT pathway plays a crucial role in defining anterior-posterior identity. The secreted element *wnt1* is expressed in the posterior-facing wounds and the secreted Wnt inhibitor, *notum*, in the anterior [22,23,24]. Inhibition of *wnt1* or *notum* during planarian regeneration produces a shift in polarity, giving rise to two-headed or two-tailed animals, respectively [22,25,26]. However, milder inhibition of the cWNT signal during posterior regeneration gives rise to the so-called tailless planarians, which are animals that close the wound without specificating a posterior midline, preventing the formation of a tail structure [23,24,25,26]. In these animals, neither a posterior nor an anterior organizer can be formed, since no *wnt1* or *notum* is expressed in the wounds [26]. Thus, we hypothesized that in planarians, organizers, and particularly the cWNT signal, could not only specify the identity but also enable the growth required to regenerate a complete tail.

In this study we identified *fzd1*, *fzd4-1,* and *lrp5/6* as cWNT signal receptors that are expressed in the “competent” cells that receive the cWNT signal and are required for posterior identity specification. In addition, we demonstrate the requirement of the cWNT signal to trigger the growth of the posterior blastema.

## 2. Material and Methods

### 2.1. Phylogenetic and Sequence Analysis

Protein lrp5/6 sequences were obtained from NCBI (Appendix A) and aligned together using MAFFT [27] with the FFT-NS-i strategy. IQ-TREE [28] was used to generate a phylogenetic tree with the generated alignment. All the web server options were used by default, with exceptions for the number of bootstrap alignments (set at 2500), the single branch test number of replicates (set at 2000), and the approximate Bayes test option (selected). Dendroscope3 v3.6.3 [29] with default parameters was used to visualize the tress. To identify the conserved domains, the NCBI web server was used (http://www.ncbi.nlm.nih.gov/Structure/cdd/wrpsb.cgi) [30].

### 2.2. Gene Cloning

*Smed-lrp5/6* fragments were cloned in pGEM-T Easy (Promega, Madison, WI, USA) and pCRII (Life Technologies, Grand Island, NY, USA) vectors for ssRNA and dsRNA synthesis, respectively. All primers used in this study are shown in Appendix A. The reported nucleotide sequence data are available in the Third Party Annotation Section of the DDBJ/ENA/GenBank databases under the accession number TPA: BK014290.

### 2.3. Animal Care

Planarians used in this study belonged to the asexual clonal strain of *Schmidtea mediterranea* BCN-10 and were maintained in PAM water as described in [31]. Animals were fed twice per week with liver to maintain the population, and the animals selected for experiments underwent starvation for one week.

### 2.4. RNAi Experiments

In vitro transcription (Roche) was used to synthesize double-stranded RNA (dsRNA) as described in [32]. Over three consecutive days per week, injections of dsRNA (3 × 32.2 nL per animal) into the digestive system were performed using a Nanoject II injector (Drummond Scientific Company, Broomall, PA, USA). Intact RNAi animals were injected for one week with 1000 ng/µL. The usual regenerative experimental RNAi inhibition consisted of two or three weeks of inhibition using 1000 ng/µL and amputations at the end of each week. In simultaneous gene-silencing experiments, the total amount of dsRNA injected in each animal was maintained constant. The new soaking protocol consisted of one week of inhibition using 2000 ng/µL, and the animals were amputated one day after the last injection. Pieces were then soaked in dsRNA diluted in PAM water with a final concentration of 1000 ng/µL. A laboratory film square piece was folded twice to generate a cross in the middle of the piece, followed by wrapping to obtain a cone. Then, the cone was seated and stacked on a Petri dish to avoid movement. A total of 12 µL of dsRNA diluted in PAM was placed in the middle of the cone. Using a brush, the planarian pieces were placed in the drop for 5 h. The folded cones were kept in a humid dark box at 20 °C to avoid evaporation. Then, the soaking protocol pieces were transferred to a Petri dish for a recovery period, and two washes with PAM water were performed during the first 15 min. The control animals were injected and/or soaked in dsRNA of *gfp*.

### 2.5. Whole-Mount In Situ Hybridization

RNA probes were synthesized in vitro using SP6 or T7 polymerase and DIG- or FITC-modified (Roche). For colorimetric whole-mount in situ hybridization (WISH), the previously described [33] protocol was followed. Animals were sacrificed with 5% N-acetyl-L-cysteine (NAC), fixed with 4% formaldehyde (FA), and permeabilized with reduction solution. For double fluorescent in situ hybridization (dFISH), the previous protocol was followed [34]. Animals were sacrificed with 7.5% NAC and fixed with 4% FA. An azide step (150 mM sodium azide for 45 min at room temperature) was added to quench the first signal probes. Nuclei were stained with DAPI (1:5000; Sigma, St-Quentin-Fallavier, France).

### 2.6. Immunohistochemistry Staining

Whole-mount immunohistochemistry staining was carried out as previously described [35]. Animals were sacrificed with 2% HCl, fixed with 4% FA and blocked in 1% bovine serum albumin (BSA) in 1× PBST × 0.3% (blocking solution) for 2 h at RT. Primary antibodies were incubated in blocking solution for 16 h rocking at 4 °C. Washes were per performed for at least 4 h, and secondary antibodies were diluted in blocking solution for 16 h rocking at 4 °C. The following antibodies were used in these experiments: mouse anti-synapsin (anti-SYNORF1, 1:50; Developmental Studies Hybridoma Bank, Iowa City, IA, USA), mouse anti-VC1 (anti-arrestin, 1:15000, kindly provided by Professor K. Watanabe) and rabbit anti-phosphohistone H3 (Ser10) (D2C8) (PH3) (1:500; Cell Signaling Technology, Leiden, Netherlands). The secondary antibodies used were Alexa 488-conjugated goat anti-mouse (1:400; Molecular Probes, Waltham, MA, USA) and Alexa 568-conjugated goat anti-rabbit (1:1000: Molecular Probes, Waltham, MA, USA). Nuclei were stained with DAPI (1:5000; Sigma).

### 2.7. Imagining and Quantification

In vivo images were obtained using Scmex 3.0 camera in a Zeiss Stemi SV 6 binocular loupe. WISH and whole-mount immunostaining images were captured with a ProgRes C3 camera from Jenoptik (Jena, TH, Germany). Cell counting of PH3+ staining was carried out by eye quantification in a previous defined area of each animal. Areas are schematically indicated in each figure. The total number of PH3+ cells was divided by the animal area. Double FISH confocal images were obtained with a Leica TCS SPE confocal microscope (Leica Microsystems, Mannhiem, BW, Germany). Representative confocal stacks for each experimental condition are shown. Images were blind analyzed and later grouped according to each genotype.

### 2.8. Single-Cell Visualization

PlanExp [36] at PlanNET [37] was used to perform t-SNE plots and gene coexpression counts with single-cell transcriptomic data [38]. Transcriptomes IDs of the used *Schmidtea mediterranea* genes are shown in each figure. Parameters were used by default.

### 2.9. Statistical Analysis and Visualization

GraphPad Prism 8 was used for statistical analysis and visualization. To compare the means of two populations, two-sided Student’s t-tests (α = 0.05) and box plots were used for statistical analysis and visualization, respectively. Box plots depicted the median, the 25th and 75th percentiles (box), and all included data points (black dots). Whiskers extend to the largest data point within the 1.5 interquartile range of the upper quartile and to the smallest data point within the 1.5 interquartile lower ranges of the quartile. To represent the percentage of the phenotype populations, heat maps were used, and in order to represent the percentage of gene presence in different cell-type populations, pie charts were used.

## 3. Results

### 3.1. Identification of an lrp5/6 Homolog That, Together with fzd1 and fzd4-2, Is Expressed in Posterior Blastemas during Regeneration

The *S. mediterranea* genome contains 9 *frizzled* (*fzd*) genes, which are grouped into three families (*fzd-1/2/7, fzd-5/8-4, and fzd4*) [39,40,41]. Two of these *fzd* appeared as good candidates to receive the cWNT signal during posterior regeneration: *fzd4-1*, which is expressed in the tip of the tail and when inhibited produces smaller tails [21], and *fzd-1/2/7* (*fzd1* from now on), whose inhibition produces a posterior head in intact planarians [40]. Whole-mount *ISH* (WISH) shows that *fzd1* is expressed in the nervous system and in the pharynx. During regeneration, it is expressed in both blastemas (Figure 1A and Appendix A). *fzd4-1* is highly expressed in the tail of intact animals, and during regeneration, it is only detected in posterior blastemas (Figure 1A and Appendix A).

The low receptor protein (LRP) is a transmembrane protein with an evolutionary conserved function as a coreceptor of the cWNT signal [42,43]. We identified three putative homologs in the *S. mediterranea* genome. Two of these were previously identified as homologs of low- and very low-density lipoprotein receptors (LDLR and VLDLR), which are closely related to LRP receptors (*Smed-ldlr* and *Smed-vldlr)* [40], and the third was not described (dd_Smed_v6_10112_0_1). The analysis of the conserved domains shows that the three homologs present low-density lipoprotein domains (Appendix A). The phylogenetic analysis demonstrates that the homolog to the LRP5/6 receptor is the new transcript, and we named it *Smed-lrp5/6* (Figure 1B). WISH shows that *Smed-lrp5/6* is expressed in the parenchyma and in the nervous system (Figure 1A). During regeneration, *Smed-lrp5/6* can be observed in the parenchyma, in the pharynx, and in the brain, but also in the posterior blastema within a duration of 48 h, suggesting its participation in the specification of the posterior fate (Figure 1A and Appendix A).

Overall, *fzd1*, *fzd4-1,* and the newly identified *Smed-lrp5/6* are expressed in posterior blastemas during regeneration, suggesting that they might have a role in defining posterior identity.

### 3.2. fzd1, fzd4-1, and Smed-lrp5/6 Specify Posterior Identity

To study the function of each receptor during posterior regeneration, we performed RNAi experiments and analyzed the posterior blastema of regenerating head and trunk fragments. Animals were injected and amputated for 3 weeks (three rounds of inhibition, see Appendix A and the Materials and Methods section). On day 7 of regeneration (7dR), after these three rounds of inhibition, most *fzd1* and *lrp5*/*6* (RNAi) regenerating heads presented a smaller and indented posterior blastema compared to controls, a phenotype related to a lack of posterior identity (Figure 2). *fzd4-1* (RNAi) regenerating heads also presented smaller posterior blastemas, but in a lower percentage (Figure 2). Regenerating trunks also presented posterior defects, although they were less penetrant (Appendix A). In the trunk fragments, we also observed that all RNAi animals showed some defects in the regenerating head, which presented smaller blastema and smaller eyes (Appendix A). The defects in the eyes were corroborated with anti-arrestin (VC-1) immunostaining, and the smaller brains were also visualized with nuclear staining (DAPI) (Appendix A). Despite the defects in the anterior regeneration, *fzd1, lrp5/6*, and *fzd4-1* (RNAi) animals presented normal expression of *notum*, a marker of the anterior pole [22,44] (Appendix A).

When cWNT is silenced, planarians present a shift in polarity or a milder phenotype that is tailless. In tailless animals, the ventral nerve cords (VNCs) do not fuse at the posterior tip, and *wnt1*, which is expressed in the posterior midline in *wild type* animals, is absent or delocalized [23,25,26]. The analysis of the VNCs through anti-synapsin (3C11) staining demonstrates that *fzd1, fzd4-1,* and *lrp5/6* (RNAi) regenerating heads were not able to close the VNCs in the posterior tip as controls (Figure 2). Anti-synapsin staining also shows that inhibition of *fzd1* and *lrp5/6* but not *fzd4-1* prevented the regeneration of a new pharynx (Figure 2). Analysis of *wnt1* by WISH showed a decrease in and a delocalization of *wnt1* in the posterior blastemas after RNAi of the three receptors, with accentuation in *fzd1* (RNAi) (Figure 2). This is a trait of tailless planarians generated after cWNT inhibition [26], which are not able to properly form the posterior midline. In agreement, expression of *slit*, a marker of the midline, was absent or disorganized in *fzd1 and lrp5/6* (RNAi) animals (Figure 2). In *fzd4* (RNAi) animals, which present a milder phenotype, *slit* expression was not affected.

Overall, these results demonstrate that *fzd1, fzd4-1,* and *lrp5/6* present a tailless phenotype, phenocopying the one described when inhibiting other elements of the cWNT pathway as *wnt1*, *wnt11-2,* or *βcat1* in planarians [23,25,26,45]. Thus, *fzd1*, *fzd4-1,* and *lrp5/6* might be the receptors activated by the cWNT signal to specify posterior identity.

### 3.3. Smed-fzd1 and Smed-lrp5/6 Cooperate in Specifying Posterior Identity

Since LRP5/6 acts as a coreceptor of Fzd to activate the cWNT signal [46], we studied their possible cooperation. First, we studied their expression in different cell types. It has been previously reported that *Smed-fzd1* and *Smed-fzd4-1* are expressed in neoblasts and muscle cells [40]. Through double FISH analysis, we demonstrate here that *lrp5/6* is also present in neoblasts and muscle cells (Appendix A). Analyzing single-cell databases [36], we found that in intact animals, *Smed-fzd1, Smed-fzd4-1,* and *Smed-lrp5/6* were present in several cell types (Appendix A), and there was coexpression between them, mostly in the epidermis, muscle cells, neurons and neoblasts (Appendix A). Since we were interested in the signaling occurring when regenerating a posterior organizer, we analyzed the expression of *wnt1*, *wnt11-2,* and the three receptors in the single-cell databases corresponding to tail fragments [36] using the Plannet interface (https://compgen.bio.ub.edu/PlanNET/planexp) [36,37]. Interestingly, we observed that very few cells coexpress *wnt1* and *wnt11-2* (Appendix A). Among the *wnt1+* or the *wnt11-2+* cells, cells expressing any combination of the three receptors were identified (Appendix A). *Smed-lrp5/6* and *Smed-fzd1* showed more similar phenotypes compared to *wnt1* (RNAi) when silenced. Thus, we hypothesized that those two receptors can cooperate in specifying posterior identity. To test this hypothesis, we simultaneously inhibited both receptors and studied regenerating heads and trunks. The results show that when silencing *Smed-lrp5/6* and *Smed-fzd1,* all knockdown head-regenerating animals present a tailless phenotype (Figure 3). However, it is in the trunk fragments where the cooperation of the two receptors is more evident. In this case, only in the double *Smed-lrp5/6* and *Smed-fzd1* RNAi condition does the animal partially show a shift in polarity and become two-headed, while the others all remain tailless (Figure 3). Immunostaining with anti-arrestin (3C11) demonstrates the appearance of the posterior head (Appendix A).

Overall, these results suggest that *fzd1* and *lrp5/6* cooperate during regeneration to specify posterior identity and could be the receptors of Wnt1.

### 3.4. The cWNT Pathway Triggers Cell Proliferation during Regeneration

Not only does the tailless phenotype observed after cWNT inhibition [25,26] show patterning defects, but blastemas also appear smaller, which could result from a decrease in the proliferative response. After amputation, two proliferative peaks are observed in planarian wounds: the first is general and appears at 6 h of regeneration (hR), while the second is local and occurs at 48 hR (hours of regeneration) [18]. To identify which role could have the cWNT signal in controlling proliferation, we analyzed the levels of PH3, which label cycling cells in the M phase [47], after *wnt1* and *wnt11-2* (RNAi). We analyzed the trunk fragments, because after *wnt1* inhibition head fragments produce a high percentage of two-headed animals [24,26], and the aim was to analyze proliferation in the tailless phenotype, which does not show any organizing activity [26]. Performing the usual protocol of RNAi inhibition (two rounds of RNAi and cut) [48], and then immunostaining with anti-PH3, we observed no significant changes in proliferation at 6 hR and an increase at 48 after *wnt1* RNAi (Appendix A). The opposite results were observed after *wnt11-2* RNAi, that is, an increase in proliferation at 6 hR but no differences at 48 h (Appendix A). According to the dynamics of expression, this result was not expected, since *wnt1* is expressed in posterior blastemas already at 6 hR [23,24]. We reasoned that since both genes were already silenced for two weeks before the analysis of the regenerating wounds, the animals could have suffered from remodeling before the last amputation, and this could have affected the regenerative process. Furthermore, 20% of the resulting animals showed a shift in polarity; thus, they were regenerating an anterior organizer in the posterior wound (Appendix A), which could also impact the proliferative response. Therefore, we designed an alternative approach to restrict the time of inhibition of the genes and ensure that none of the animals suffered from a shift in polarity. In this case, we injected dsRNA for three consecutive days using a higher concentration of dsRNA, and after amputation, animals were soaked for 6 h in dsRNA diluted in PAM water (Figure 4A and Appendix A) (see the Materials and Methods Section for the soaking protocol). When using this protocol and analyzing PH3 levels, we found that both *wnt1* and *wnt11-2* (RNAi) animals show a reduction on the first mitotic peak, while the second is increased in *wnt1* and not affected in *wnt11-2* (RNAi) animals (Figure 4A).

The same soaking protocol was then used to analyze the proliferative response after inhibition of *lrp5/6*. In this case, the regenerating heads were analyzed since they showed more penetrance of the tailless phenotype than the trunk fragments (see Figure 2 and Appendix A). *lrp5/6* (RNAi) animals showed a reduction in the first peak, although it was not statistically significant, and all differences were observed in the second peak in comparison to controls (Figure 4B and Appendix A).

To test whether the cWNT signal could be also controlling proliferation during planarian homeostasis, we analyzed PH3+ cells in *wnt1* (RNAi)-intact animals. After one week of inhibition, the animals did not present the in vivo phenotype (Appendix A). However, the proportion of mitotic cells in the posterior region, where *wnt1* is expressed, was decreased when compared to controls (Appendix A), suggesting that *wnt1* controls proliferation in homeostatic conditions.

Overall, our results show that cWNT inhibition produces not only patterning defects but also a decrease in proliferative cells. Furthermore, our data highlight the impact of the strategy used to modulate gene expression when early time points of regeneration are studied.

## 4. Discussion

To specify posterior identity and regenerate a proper tail, the appearance of a posterior organizer, which is composed of muscular *wnt1+* cells, is required during the first hours of regeneration in planarians [23,24]. It is known that the subsequent activation of other posterior Wnts (mainly *wnt11-2*) is required to pattern the tail [24,25]. However, the receptors of the cWNT signal have not been carefully examined in the currently available literature. Here, we have identified *fzd1*, *fzd4-1,* and *lrp5/6* as receptors that mediate the cWNT action during posterior regeneration. According to our results, Fzd1, rather than Fzd4-1, appears to be the direct receptor of the Wnt1 signal from the organizing cells, since its RNAi produces a stronger and more penetrant phenotype during posterior regeneration. Furthermore, *fzd1* expression in posterior wounds is detected earlier than that of *fz4-1,* indicating that it could be receiving an earlier signal of Wnt1, which could also account for the stronger phenotype observed after its inhibition. Our data also indicate that LRP5/6 could act as a coreceptor of Fzd1 since its inhibition produces a very similar phenotype regarding the posterior midline disorganization and the inability to regenerate a new pharynx. The absence of a pharynx in *fzd1*and *lrp5/6* RNAi animals could result from their early role in posterior specification [49] or from their direct role in regeneration of the pharynx, since they are both expressed in it. The finding that simultaneous inhibition of *fzd1* and *lrp5/6* produces a synergistic effect and that it can produce a shift in polarity, also shown in *wnt1* (RNAi) animals, further supports the notion that LRP5/6 acts as an Fzd1 coreceptor. Thus, an Fz1/LRP5-6 signal would be evolutionarily conserved in receiving the cWNT signal as described in other models [50,51,52,53]. Fzd4-1 could act as the receptor of Wnt11-2, since its expression pattern and RNAi phenotype are very similar, always producing tailless animals, and it could also act as a receptor of the late Wnt1 signal (Figure 5).

Interestingly, despite the fact that the Fdz1/LRP5-6 receptors seem to mediate the signal from the organizer, they are not expressed specifically in posterior wounds, but *fzd4-1* is the receptor found specifically in posterior-facing wounds. According to our data and others [21,23,25,26], we propose that, during posterior regeneration, the early Wnt1 interacts with the Fzd1/LRP5-6 receptors in the nearby cells, and, then, in these cells, the expression of *wnt11-2* and *fzd4-1* is activated, generating a broader field of cells around the posterior organizer that corresponds to the “competent” field of cells responding and executing the posterior program (*wnt11-2+* and *fzd4-1+*). In these cells, Fzd4-1 may be the receptor of Wnt11-2 or the late Wnt1 signal (Figure 5). We have found that the three receptors are coexpressed with *wnt1+* or *wnt11-2*+ cells, at least in intact tails, suggesting that both cell types may be activating their own transcription, thereby establishing a robust posterior program (Figure 5). Using the cell-sequencing database, we also found that *wnt1* and *wnt11-2* do not coexpress but can be identified as two different cell populations, which in the proposed model would correspond to the organizing and receiving cells, respectively (Figure 5).

The Wnt1-Fz1/LRP5-6 signal appears not only necessary to specify the posterior fate but also to trigger the proliferative response required for proper regeneration. This is not surprising since the requirement of Wnt-Fzd-LRP signaling to mediate proliferation has been extensively reported in other stem-cell-based systems, such as in intestinal crypts [54], among others [55]. In acoels, it was recently demonstrated that inhibition of *wnt3*, which is expressed in posterior muscular cells [56], also affected stem cell proliferation [57], although in this case, the receptors remain unknown.

The inhibition of *wnt1*, *wnt11-2,* or *lrp5/6* affected the proliferation of neoblasts at 6 h of regeneration, but not at 48 hR. It must be noted that the mitotic response that occurs at 6 hR is general, and that at 48 hR is localized in the wounds. Thus, this result means that inhibition of the function of the posterior organizer has a very early role in the proliferative response of the stem cells, but it does not affect the later mitotic peak, which is responsible for the formation of the blastema observed in tailless animals. Since during planarian regeneration a very early apoptotic peak of apoptosis appears locally in the wounds (4 hR) [19], interfering with the function of the organizer may directly affect the apoptotic response and, by extension, proliferation. In other systems, it has been reported that dying cells generate cWNT ligands that modulate stem cell proliferation, as in *Hydra* [58], in epithelial tissue in zebrafish [59], and in *Drosophila* imaginal discs [60,61] (also reviewed in [62]). In *Nematostella*, it has also been proposed that cell death might trigger regeneration and proliferation [63]. Thus, the analysis of cell death in *wnt1* (RNAi) animals may help in clarifying the role of the organizer with respect to cell death and cell proliferation.

A further aspect to note regarding the role of the organizer in the control of proliferation is the finding that the results were highly dependent on the protocol used for gene inhibition. This observation must be considered for further studies on early events in regeneration. When the protocol of gene inhibition takes several days, which is usually the case, planarians begin to remodel according to the new signals, and this can influence the early regenerative response. This observation is related to the finding that inhibition of *wnt1* in homeostatic animals also produces a decrease in the proliferative rates of the cells nearby. The *wnt1+* cells in the posterior midline of adult animals do not maintain the identity, since a shift in the polarity during homeostasis is observed just after *βcat1* inhibition [45], but they do have a role in maintaining the homeostasis of posterior tissues.

## 5. Conclusions

The formation of organizers or organizing centers is a conserved mechanism in evolution to pattern a growing field of cells. This continuously occurs during embryonic development, but it is also required during regeneration of new structures in adult animals. In highly regenerative animals, such as *Hydra*, planarians, or zebrafish, the existence of regenerative organizers has been proved, and the molecular mechanism underlying their properties is currently being deciphered. Through the study of these plastic models, it has been shown that the cWNT signal appears as an evolutionary conserved signal providing organizing properties. The identification of the cells in the organizer and the factors associated with them, expressed both in the organizer itself and in the surrounding cells, provides us with a new frame to understand regeneration. A crucial property of organizers is that they integrate patterning and growth, which are both the basis of a successful regenerative process. We envisage that mirroring the knowledge gained through the study of organizers in animal models to mammals and humans will provide a new frame to further explore the field of regenerative medicine.

## Figures and Tables

**Figure 1 genes-12-00101-f001:**
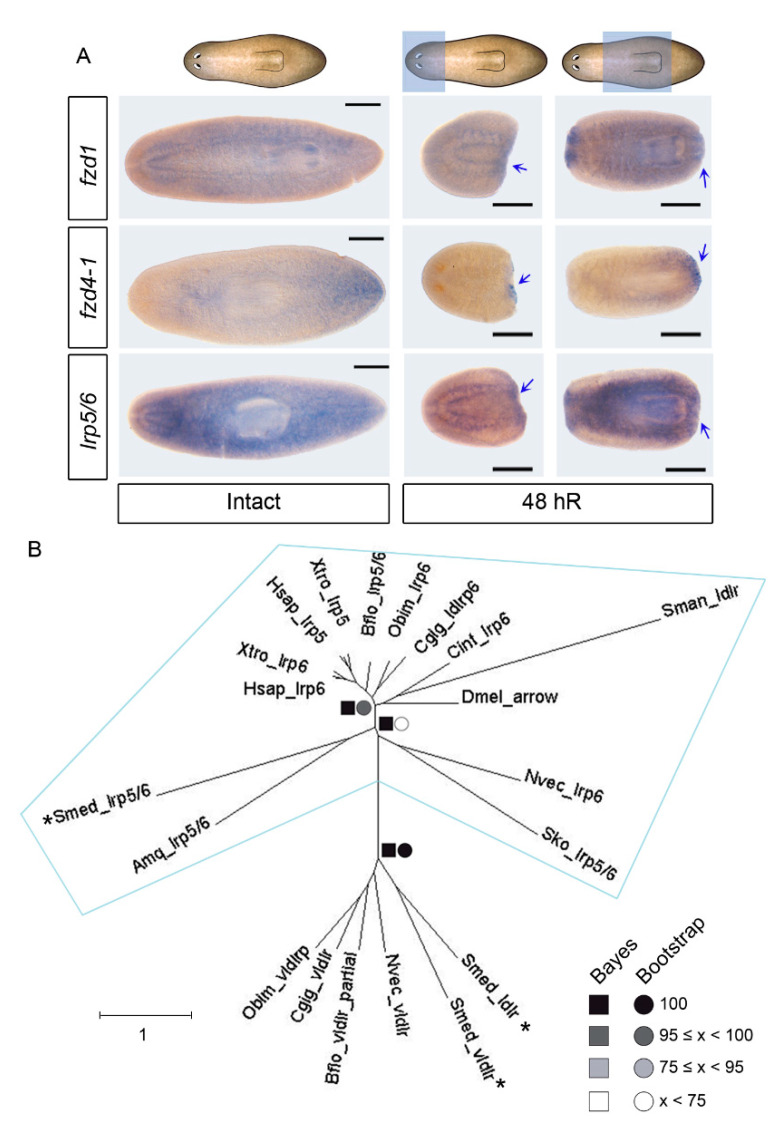
*fzd1*, *fzd4-1,* and *lrp5/6* are expressed in posteriors blastemas after amputation. (**A**) Whole-mount in situ hybridization (WISH) of *fz1*, *fzd4-1,* and *lrp5/6* in intact animals and at 48 h of regeneration (48 hR), showing its presence in posterior blastemas (blue arrows). Blue shadows in the schematic cartoons represent the studied regenerative pieces: heads and trunks. Scale bar: 100 µm. (**B**) The phylogenetic tree based on LRP sequences (Appendix A) showed that *lrp5, lrp6,* and *lrp5/6* genes cluster together. At nodes, values for the approximate Bayes (square) and Likelihood (circle) ratio tests are shown. Colour indicates % of confidence. Dark asterisks indicate *Schmidtea mediterranea* (*Smed*) genes. Scale indicates expected aminoacidic substitution per site. The following species were used: *Homo sapiens* (*Hsap*), *Xenopus tropicals* (*Xtro*), *Branchiostoma floridae* (*Bflo*), *Saccoglossus kowalewski (Sko*), *Crassostrea gigas* (*Cgig*), *Octopus bimaculoides* (*Obim*), *Schistosoma mansoni* (*Sman*), *Nematostella vectensis* (*Nvec*), and *Amphimedon queenslandica* (*Amq*).

**Figure 2 genes-12-00101-f002:**
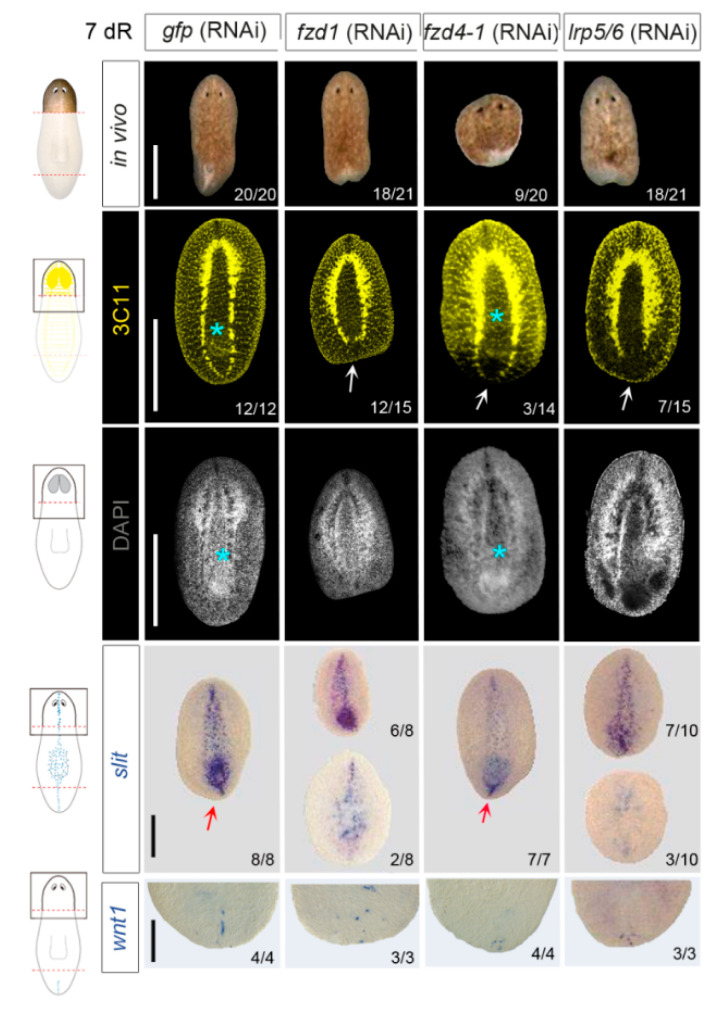
*fzd1, fzd4-*1, and *lrp5/*6 inhibition produces tailless heads. In vivo images of *fzd1*, *fzd4-1,* and *lrp5/6* (RNAi) regenerating heads showed indented or rounded blastemas. Immunostaining of the neural system using α-SYNAPSIN (3C11) showed that *fzd1*, *fzd4-1,* and *lrp5/6* (RNAi) animals do not properly close the ventral nerve chords in the posterior tip (white arrows) and that the new pharynx is not regenerated after *fzd1* and *lrp5/6* RNAi (cyan asterisks), which is corroborated with nuclear staining (DAPI). WISH using a *slit* probe shows the disorganization of the midline (red arrows) in *fzd1* and *lrp5/6* (RNAi) animals. WISH of *wnt1* in regenerating blastemas shows its delocalization after the inhibition of the three receptors. On the left, schematic illustrations are added showing which parts were studied (dark boxes) after amputation (red dashed line) and the expression of 3C11, *slit,* and *wnt1* in intact animals. Scale bars: 200 µM.

**Figure 3 genes-12-00101-f003:**
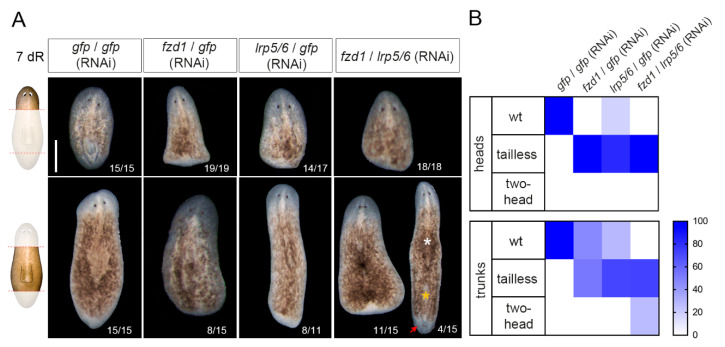
The simultaneous inhibition of *fzd1* and *lrp5/6* leads to changes in posterior polarity. (**A**) In vivo images of *fzd1, lrp5/6,* and the double RNAi showed that compared to controls, heads presented tailless, rounded blastemas. Trunks of *fzd*1 or *lrp5/6* (RNAi) also presented the tailless phenotype, and the double RNAi presented tailless and two-head planarians (red arrow). Asterisks indicate the old (white) and new (orange) pharynges. Schematic illustrations are added showing which parts were analyzed. Scale bar: 500 µm. (**B**) Heat maps show the percentage of phenotypes observed in each experimental condition.

**Figure 4 genes-12-00101-f004:**
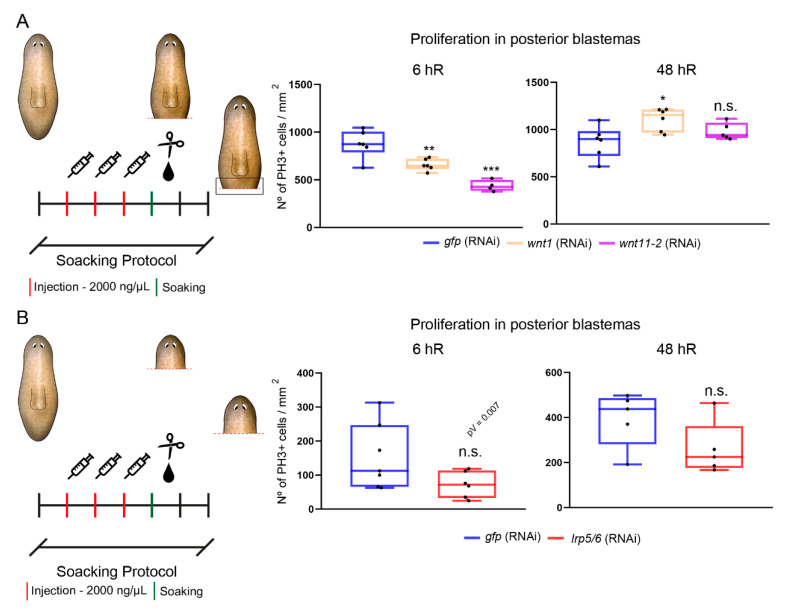
*wnt1, wnt11-2,* and *lrp5/6* (RNAi) animals show proliferative changes during posterior regeneration. (**A**) Schematic illustration indicating the RNAi procedure using the soaking protocol. The quantified area and the amputated area of the trunks are indicated with a dark box and dashed red line, respectively. Quantification of PH3+ cells after the soaking protocol at 6 h of regeneration (hR) (*gfp*, *n* = 6; *wnt1*, *n* = 6; *wnt11-2*, *n* = 5; ** *p* < 0.01, *** *p* < 0.001) and 48 hR (*gfp*, *n* = 6; *wnt1*, *n* = 6; *wnt11-2*, *n* = 5; * *p* < 0.05, n.s.). (**B**) Schematic illustration indicating the RNAi procedure using the soaking protocol. The quantified area and the amputated area of the heads are indicated with a dark box and dashed red line, respectively. Quantification of PH3+ cells after the soaking protocol at 6 h of regeneration (hR) (*gfp*, *n* = 7; lrp5/6, *n* = 6; n.s.) and 48 hR (*gfp*, *n* = 5; lrp5/6, *n* = 5; n.s.).

**Figure 5 genes-12-00101-f005:**
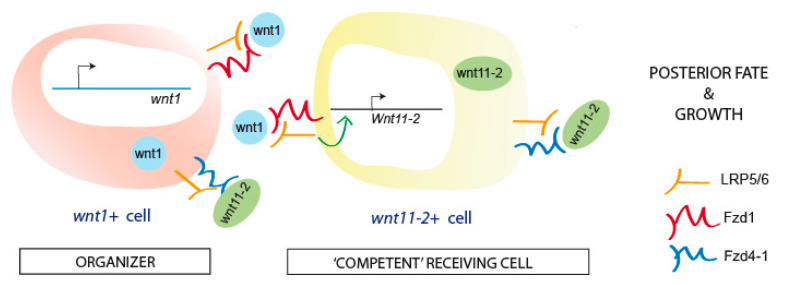
Model of how FZD and LRP5/6 receptors could be receiving the cWNT signal in the organizer (*wnt1+*) and in the receiving cells (*wnt11-2+*) to specify the fate and promote growth of the posterior wound during planarians regeneration.

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
