# Peer review of "WNT-FRIZZLED-LRP5/6 Signaling Mediates Posterior Fate and Proliferation during Planarian Regeneration"

_genes, 2021, doi:10.3390/genes12010101_

Round 1

Reviewer 1 Report

Planarians exhibit the remarkable ability to regenerate a fully functional head and tail within 10 to 14 days after amputation. Anterior and posterior poles that possess putative organizer activity coordinate the patterning and growth of the regenerating head and tail, respectively. The posterior pole is characterized by the expression of the secreted signaling molecule wnt1, however currently very little is known how wnt1 acts onto surrounding cells to mediate patterning and growth of the regenerating tissue. In the manuscript by Pascual-Carreras the authors reach out to tackle this question and report on Wnt receptors that receive the wnt1 signal from posterior pole cells. The authors propose that wnt1 is released from posterior pole cells and acts on surrounding “competent” receiving cells through fzd-1 and lrp5/6. This in turn could then lead to the expression of wnt11-2 that might signal through fzd-4 and lrp5/6.

Overall, the authors shed light on an important question, which is how an “organizer” works and I am convinced that this manuscript will be of great interest for the readership of Genes. However, before reaching a final decision some additional experiments are required to strengthen the presented results as well as the proposed model. My major concerns are listed below.

Major concerns

The authors propose that fzd1 and lrp5/6 could be the receptors of wnt1 and that fzd-4 and lrp5/6 could be the receptors of wnt11-2 in competent receiving cells. Please knockdown wnt1 together with lrp5/6 and fzd1 but also fzd4-1. Does e.g. the knockdown of wnt1 together with lrp5/6 enhance the wnt1 phenotype? Does the knockdown of wnt11-2 together with e.g. lrp5/6 enhance the wnt11-2 phenotype? Please also provide FISH data, showing that wnt1 is not co-expressed with lrp5/6, fzd-4 and fzd1 in the posterior pole “organizer”. What are the wnt1 receptors in the organizer that maintain the expression of wnt1 in the “organizer”? FISH data should also be provided to demonstrate that wnt11-2 is co-expressed with fzd-4, fzd-1 and lrp5/6 in receiving cells. In addition, the authors show that the silencing of lrp5/6 together with fzd1 leads to the regeneration of a head at posterior facing wounds. However, is such a double-headed phenotype also observed after the silencing of lrp5/6 together with fzd4-1?

The authors analyze PH3-positive cells after silencing wnt1, wnt11-2 and lrp5/6 through a combination of injection and soaking. Please provide knockdown efficiencies, showing that soaking enhances the knockdown efficiency. Lrp5/6 RNAi animals do not show a reduction in the number of PH3-positive cells. Why do the authors not analyze PH3-positive cells after the silencing of lrp5/6 together with wnt1 or wnt11-2? This would further strengthen their results that these Wnts act through this receptors to regulate cell proliferation and thus growth.  

How similar is Smed-lrp5/6 to Smed-ldlr and Smed-vldlr. Do these proteins have certain domains in common? A protein comparison at the architecture of the domain level should be presented.

Minor concerns

Did the authors test for the expression of fzd1, fzd4-1 and lrp5/6 in regenerating tails? I suggest showing the expressing of fzd1, fzd4-1 and lrp5/6 in head, trunk and tail fragments at 48 hpa together with intact animals in main Figure 1. The phylogenetic tree could be moved to the supplement. Please also add arrows that point towards the expression, which will help readers’ non-familiar with planarians.

Line 166-168: “Whole mount ISH shows that fzd1 is expressed in both blastemas and fzd4-1 is specifically expressed in posterior blastemas as soon as 24h after amputation”. The 12 and 24 hpa time points in Supplementary Figure S1 do not look very different. Please either provide high magnification images or add arrows that point towards the expression at 24 hpa.

Line 187-188: “WISH shows that Smed-lrp5/6 is expressed in the parenchyma and in the nervous system”. It seems that Smed-lrp5/6 but also fzd1 are expressed in the regenerating pharynx. Please clarify!

Supplementary Figure S2B and S1: From the ISHs it seems that fzd4-1 is not expressed during head regeneration. How do the authors explain the head regeneration defect of fzd4-1 RNAi animals? What could be the Wnt that signals through fzd4-1, fzd1 and lrp5/6 in head regenerating animals?

Figure 2: Please add arrows that point towards the tail regeneration defect and towards the VNCs that do not close.

The authors show a decrease and delocalization of Wnt1 in Figure 2. Is the expression of Wnt1 reduced or elongated in fzd1 RNAi animals?

Figure S3A: Please provide high magnification images for double positive cells and point with an arrow towards a double positive cell.

Figure S3A and S4 (DAPI channel) miss scale bars.

The title is a bit awkward to read. I suggest changing the title to “Wnt-Frizzled-LRP5/6 signaling mediates posterior fate and proliferation during planarian regeneration”.

Line 26: What do the authors mean with Fzd1-LRP5/6 axis?

Line 202: “…could also observe that all RNAis presented…”. RNA interference is a technique. This sentence should read “…could also observe that all RNAi animals showed…”

Supplementary Figure S2A: “3dr Round” should read “3rd Round”.

Tailless, Two-headed, Organizer, Co-expression should be written in lowercase throughout the text.

Line 335: lrp5-5 should read lrp5-6.

Line 16: “…that secrete an extracellular protein…” I assume that this is not what the authors intend to say, as an organizer releases many secreted proteins and not only one.

Hydra should be written italicized.

Line 74: “…no posterior either anterior organizer can be formed…” should read “…neither a posterior nor an anterior organizer can be formed…”

Line 118 should read: RNA probes were synthesized in vitro…

Line 126 should read: …was carried out as previously described.

Line 130 should read: …the following antibodies were used…

Line 162 should read: The S. mediterranea genome…

Line 185 should read: …was not described…

Line 234 should read: For that reason…

Line 258 should read: …general that appears…

Line 273 should read: …soaked for…

Reviewer 2 Report

The authors studied the roles of several wnt receptors and co-receptors in planarian regeneration. This is an additional contribution to the many contributions that the senior authors have made to this field. The paper identifies interesting phenotypes, in particular the fz1/lrp5/6 (RNAi). Imaging and quality should be improved, and more effort should be made to explain the phenotypes considering the extensive body of work on anterior-posterior patterning in planarian regeneration. 

Major comments:

  1. The authors show that fzd1, fzd4-1, and lrp5/6 RNAi animals have major defects in anterior regeneration, which is very important for understanding the function of these genes, and more analysis is required. For example, do RNAi animals generate an anterior pole? Do they have pole defects that are similar to those that are described in the bottom panel of figure 2? fzd1 and lrp5/6 are expressed in the planarian brain, and therefore could have roles in planarian brain regeneration and homeostasis. A more thorough analysis of these phenotypes is encouraged.
  2. Tailless phenotype: It is unclear whether the animals do not develop a tail or fail to completely specify a posterior identity. Do the RNAi animals show a normal wound-induced gene expression (e.g., posterior wnt).
  3. Fig S3A: Figure quality is insufficient. Authors should replace the image with a larger image and include an additional image that is higher resolution. Moreover, channels should be separated to allow examination of co-localization and a scale bar has to be included. Analysis shown in Fig S3 is not in agreement with FISH analysis done by Witchley et al., Cell Rep, 2013, showing that over 85% of the fz-4+ cells are muscle cells.
  4. Fig 4: representative PH3-labeling images should be shown with description of the cell counting. The result should be better explained. How come a blastema is not formed (tailless phenotype) yet cell proliferation is not significantly affected? Is that a 48 hr post amputation specific effect? If indeed, when do the control and lrp5/6 (RNAi) mitotic rates diverge? Also, I do not see why the wnt11-2 results are included. They appear irrelevant and out of place.
  5. Does the injection-soaking RNAi protocol produce phenotypes in similar severity and penetrance, when compared to the initial RNAi procedure that the authors have used? If it does not then how applicable are these results for the analysis of the phenotype shown in figure 2?
  6. Line 310: fz4 has been studied in several planarian papers.

Minor comments:

  1. Editing is required. Language is often colloquial and grammatically incorrect. Two examples of many: lines 195-196: Animals were injected and cut during 3 weeks that is what we called 3 rounds of inhibition. Line 185: “the third one was no described.
  2. References should be checked and reformatted (e.g., ref 40).
  3. Fig S3B: What is the scale of gene expression? This is not defined in the figure.
  4. The result shown in figure 3 for the double RNAi is intriguing, and is distinct from experiments performed by Petersen and Reddien (PNAS, 2009) on the Wnt pathway, which showed that a double-headed phenotype following wntP-1 (RNAi) is more penetrant in head fragments. Could the authors speculate on potential reasons for this difference?

Round 2

Reviewer 1 Report

Line 45 typing mistake: ...using adult s. should read using adult Hydra.

The authors have addressed most of my comments.